# Synthesis and Activity of Ionic Antioxidant-Functionalized PAMAMs and PPIs Dendrimers

**DOI:** 10.3390/polym14173513

**Published:** 2022-08-27

**Authors:** Katia Bacha, Julien Estager, Sylvie Brassart-Pasco, Catherine Chemotti, Antony E. Fernandes, Jean-Pierre Mbakidi, Magali Deleu, Sandrine Bouquillon

**Affiliations:** 1Molecular Chemistry Reims Institute UMR CNRS 7312, Reims Champagne-Ardenne University, Boîte n° 44, B.P. 1039, 51687 Reims, France; 2Laboratory of Molecular Biophysics at Interfaces (LBMI), Gembloux Agro-Bio Tech-Liège University Passage des Déportés, 5030 Gembloux, Belgium; 3Certech, Rue Jules Bordet, 45 Zone Industrielle C, 7180 Seneffe, Belgium; 4UMR CNRS/URCA 7369 (MEDyC), Reims Champagne Ardenne University, 51 Rue Cognacq Jay CS30018, 51095 Reims, France

**Keywords:** dendrimers, phenolic acids, antioxidant, microwave, flow chemistry

## Abstract

For this study, new dendrimers were prepared from poly(propylene imine) (PPI) and polyamidoamine (PAMAM) dendrimers using an efficient acid-base reaction with various phenolic acids. The syntheses were also optimized in both microwave and microfluidic reactors. These ionic and hydrophilic dendrimers were fully characterized and showed excellent antioxidant properties. Their cytotoxic properties have been also determined in the case of fibroblast dermal cells.

## 1. Introduction

Dendrimers are hyperbranched macromolecules having, in general, a three-dimensional, monodisperse and globular structure, most of the time with a perfect tree-like structure [1,2]. They are largely studied mainly for their encapsulation capacity, particularly in the biomedical field [3]. Recently, Kaur et al. presented an interesting review concerning a comparative study of poly(propyleneimine) (PPI) and polyamidoamine (PAMAM) dendrimers, some of the most-used dendrimers [4]. In recent years, we have developed various dendrimers containing biobased moieties to make these compounds more eco-compatible. These have been used in catalysis, medical imaging and depollution [5,6,7,8]. The present study is devoted to the preparation of new ionic dendrimers that present, moreover, antioxidant properties. Indeed, these new macromolecules are designed for original use as carriers in cosmetics; the encapsulation will limit active compound degradation, their ionic character will improve their solubility in compatible solvents for cosmetics, and their antioxidant power will protect the macromolecules themselves and reinforce their beneficial encapsulating role. To our knowledge, no dendrimers presenting both ionic and antioxidant characteristics have yet been described in the literature.

However, the synthesis of ionic dendrimers has previously been described in the literature, for example, dendronized pyridinium units surrounded by a shell of carboxylic acids [8] or water-soluble polyanionic dendrimers, presenting great potential applications as antiviral drugs [9]. Ionic dendrimers have also been used as a base for polyamide membranes for ion separation [10] or electrolytes for LiS batteries [11]. Their use in the form of liquid crystals has also been investigated alongside, for instance, ionic dendrimers as poly(ethylene imine) polymers functionalized by oxadiazole [12,13] or fluorinated and perhydrogenated chains [14].

Antioxidant dendrimers have been prepared using two approaches. The first approach consists of the synthesis of new dendritic structures presenting antioxidant properties; the second approach is based on the capacity of existing dendrimers to encapsulate antioxidant compounds. In the first approach, phenols could be used as the starting material of choice to build antioxidant moieties. Indeed, dendritic polyphenols have been prepared and evaluated as antioxidant compounds [15,16,17], for use either coupled with cis-platin as an anticancer agent [18] or mostly in polyolefin to avoid the peroxidation of the olefins [19,20]. Carbosilane, when functionalized by polyphenols, also presents antioxidant, antibacterial and anticancer properties [21,22]. The decoration of PAMAM [23,24,25], glycodendrimers [26] or original dendrimers (triazole-bridged, hyperbranched polyurethanes, with an enone or anthraquinone or melamine core) [27,28,29,30,31,32]; these are partnered with specific entities, such as polyphenols, Meldrum’s acid derivatives, carbazole, fluorescein or Rhodamine that have also led to new dendritic molecules presenting, among others, good antioxidant properties. Decorated peptide dendrimers have also presented excellent antioxidant properties [33], as well as original radical dendrimers [34].

The second approach for preparing antioxidant dendrimers is related to their capacity to encapsulate various types of compounds (organic or organometallic compounds, salts and nanoparticles). Indeed, dendrimers or dendrimer nanoparticles have largely been used in therapeutic domains. Curcumin [35,36], gold nanoparticles [37], astragaloside [38], cis-platin [39] and sinomenine or minocycline [40,41] were encapsulated in different dendrimers (PAMAM, modified PAMAM and carbosilane dendrimers), leading to controlled release for targeted therapy. Gallic acid-modified PAMAM dendrimers were also explored as novel strategies to fight the chemoresistance of tumor cells [42,43]. These last compounds were also used as devices for the long-term preservation of essential oils [44]. Dendrimers such as PAMAM [45], chitosan-poly(amidoamine) dendrimer [46], or dendrimer-like glucan [47] were also employed to stabilize antioxidants, such as carotene [47], coenzyme Q10 [48], resveratrol [49] or polyphenols [50], showing excellent antioxidant activities.

In the present study, we will describe the preparation of new ionic dendritic structures from a simple acid-base reaction between PAMAMs or PPIs and phenolic acids without a coupling agent. The nitrogen-based moieties are cationic, and the anion is the carboxylate form of the phenolic acids. The antioxidant properties are carried by ionic species; the electrostatic linkage can enhance the compounds’ solubility in water and may influence the cytotoxic properties of the dendritic structures. These compounds have been synthesized in a swift and efficient way, using either microwave or microfluidic reactors. Both techniques—which are strongly linked to sustainable chemistry [23,51,52]—are known to significantly accelerate the reaction kinetics, taking advantage of either very fast and homogeneous heating through dielectric loss, in the case of microwaves (MW) [53], or from very efficient mass and heat transfer, in the case of flow chemistry [54]. These techniques have been used for the synthesis of many chemicals, including, for example, active pharmaceutical ingredients [55] or biobased compounds [56,57].

## 2. Results and Discussion

### 2.1. Synthesis of the Ionic Dendrimers

In this study, oligomers synthesized from 3 generations of PPI dendrimers and 3 biobased phenolic acids (phloretic, ferulic and caffeic acids) were first targeted.

Poly(propylene imine) (PPI) dendrimers [58] are also known as DAB-Am-x dendrimers (DAB, for the diaminobutane core, and x = 4, 8, or 16, for the number of primary amine end groups associated with generations 1, 2, or 3, respectively). Phloretic acid (PhA), found, among others, in olives [59] or in the bran extracts of traditional rice cultivars [60], was used in a recent application as a biocompatibilizer for immiscible polymer blends [61]. Ferulic acid (FA) is widely present in plants in either its free or conjugated forms and is well known for many various applications in the medical field, thanks to its bioactivity, for the treatment of cancer, diabetes, and lung and cardiovascular diseases [62]. Ferulic acid has a protective role for the main skin structures (keratinocytes, fibroblasts, collagen and elastin); it inhibits melanogenesis, enhances angiogenesis, and accelerates wound healing [63]. Caffeic acid (CA), extracted from specific plants’ biomass, is a dietary polyphenol that is also considered a promising drug candidate, thanks to its efficiency against inflammatory, neurodegenerative, oncologic and metabolic disorders [64,65]. Caffeic acid also presents a photoprotective effect in irradiated lymphocytes [66] and could act as a potent chemopreventive agent against skin cancer [67].

The synthesis of these new families of functionalized PPIs with the phenolic acid derivatives, PPIs-PhA, PPIs-FA and PPIs-CA, was realized by an acid-base reaction of commercial PPIs with ferulic acid, phloretic acid and caffeic acid, respectively, in methanol or ethanol under various conditions (Figure 1, Table 1). This reaction has the advantage of not requiring coupling agents when compared to the traditional amidation reaction, for instance.

The first-generation dendrimers, PPI-1-PhA and PPI-1-FA, were obtained by a reaction between the terminal primary amines of first-generation PPI (PPI-1) with phloretic acid or ferulic acid, respectively, in refluxing methanol for 5 h (Figure 1(a)). Reflux was necessary to obtain higher yields in lower times. This reaction was confirmed by ^1^H NMR, focusing on the chemical shift of the protons of the CH_2_ groups linked to the terminal primary amine functions (2.67 ppm for PPI-1, 2.91 ppm and 2.98 ppm, respectively, for PPI-1-PhA and PPI-1-FA in CD_3_OD) and by ^13^C NMR analyses, focusing on the chemical shift of the carbon from the carbonyl group (176.3 and 169.6 ppm, respectively), for phloretic acid and ferulic acid; 180.4 and 174.7 ppm, respectively, for PPI-1-PhA and PPI-1-FA in CD_3_OD (see Section 3.1). These shifts toward lower fields are consistent with salt formation between the amines and carboxylic acids [68].

Two-dimensional NMR analyses were also performed to prove that no formation of amide functions occurred instead of forming ionic compounds. Next, the disappearance of the strong ν(CO) bands of phloretic acid and ferulic acid using IR (1699 cm^−1^ and 1689 cm^−1^, respectively), as well as the two new ν(CO) bands of ionized acids at 1540 cm^−1^ and 1396 cm^−1^ for PPI-1-PhA and at 1539 cm^−1^ and 1394 cm^−1^ for PPI-1-FA, confirmed the formation of the ionic dendrimers [68]. The purification of PPI-1-PhA and PPI-1-FA was performed by solubilizing the crude mixture in water, followed by extraction with ethyl acetate. Similarly, the second and third generations of PPIs-PhA and PPIs-FA were obtained in quantitative yields.

Concerning the preparation of PPIs-CA, the conditions were adapted because the mixture of the two reagents (PPI and caffeic acid) in methanol precipitates as a brown wax. First-generation PPI-1-CA was, therefore, obtained by reacting the terminal primary amines of a PPI-1 with caffeic acid in ethanol at room temperature under orbital stirring (280 rpm), with the dropwise addition of the caffeic acid ethanolic solution (Figure 1 (c)). The formation of PPI-1-CA was investigated by NMR, focusing firstly on the chemical shift of the H5 protons (see Section 3.1) corresponding to the CH_2_ groups that are linked to the terminal primary amine functions (2.53 ppm for PPI-1 and 2.73 ppm for PPI-1-CA in (CD_3_)_2_SO), and secondly on the chemical shift of the C6 carbons (see Section 3.1) corresponding to the carbonyl groups (168.4 ppm for caffeic acid and 171.6 ppm for PPI-1-CA in (CD_3_)_2_SO). IR analyses confirmed the nature of the product through the disappearance of the strong νCO band of caffeic acid at 1640 cm^−1^ and the appearance of two νCO bands of ionized acid around 1507 cm^−1^ and 1372 cm^−1^ for PPI-1-CA, as previously described [68]. The purification of PPI-1-CA was finally performed by washing the resulting precipitate with ethanol, followed by centrifugation. The second and third generations (PPI-2-CA and PPI-3-CA) were prepared following the same process and were obtained at yields above 90%.

The synthesis of PAMAM dendrimers functionalized with phenolic acids, PAMAMs-PhA, PAMAMs-FA and PAMAMs-CA, were performed in the same way via the acid-base reaction of commercial PAMAMs with ferulic, phloretic and caffeic acids, respectively, in methanol or ethanol (Figure 2, Table 2). However, it was necessary to increase the number of equivalents of phenolic acid (up to 1.5 equiv. per NH_2_) and the reaction time (up to 16 or 20 h), probably because of the steric hindrance caused by the morphology of the PAMAMs, which prevents the insertion of phenolic acids and the formation of ionic bonds.

The formation of PAMAM-1-PhA and PAMAM-1-FA was highlighted by ^1^H NMR, focusing on the chemical shift of CH_2_ groups linked to the terminal primary amine functions (2.75 ppm for PAMAM-1, 2.99 ppm and 3.1 ppm, respectively, for PAMAM-1-PhA and PAMAM-1-FA in CD_3_OD). The ^13^C NMR also shows a chemical shift of the carbonyl carbons (176.3 ppm and 169.6 ppm, respectively, for phloretic acid and ferulic acid; 180.6 ppm and 174.6 ppm, respectively, for PAMAM-1-PhA and PAMAM-1-FA in CD_3_OD (see Section 3.1)). Following the same procedure, the second and third generations of PAMAMs-PhA and PAMAMs-FA were prepared, with yields above 90%.

Once again, solubility issues occurred with caffeic acid; the formation of the first-generation PAMAM-1-CA was performed by reacting PAMAM-1 with caffeic acid in ethanol (instead of methanol) at room temperature, adding the caffeic acid solution dropwise (Figure 2 (b)). In this case, no heating of the medium was required to obtain good yields. The formation of PAMAM-1-CA was followed by ^1^H and ^13^C NMR, focusing on the chemical shift of the H11 protons corresponding to the CH_2_ groups linked to the terminal primary amine functions (see Section 3.1) (2.55 ppm for PAMAM-1 and 2.78 ppm for PAMAM-1-CA in (CD_3_)_2_SO) and on the chemical shift of the C12 carbons corresponding to the carbonyl function (see Section 3.1) (168.4 ppm for caffeic acid, 172.5 ppm for PAMAM-1-CA in (CD_3_)_2_SO). The disappearance of a strong νCO band of caffeic acid, observed in IR at 1640 cm^−1^, and the appearance of two νCO bands of ionized acid around 1522 at 1580 cm^−1^ and 1376 cm^−1^ for PAMAM-1-CA, confirm the formation of the ionic substrates. The second and third generations (PAMAM-2-CA and PAMAM-3-CA) were prepared similarly, with good yields of higher than 99%.

### 2.2. Synthesis in MW and Microfluidic Reactors

The feasibility of the reaction using microwave irradiation was demonstrated with PPI dendrimers, when the different syntheses were then optimized. Taking advantage of the fast and efficient heating induced by microwave organic synthesis, the formation of PPIs-PhA, PPIs-FA and PPIs-CA could be achieved in only 5 min at 100 °C (Figure 1 (b)) with very high isolated yields. The different results obtained are summarized in Table 3.

Flow synthesis in a mesofluidic reactor was also investigated in the search for a process that is both time/energy-efficient and easily scalable. Considering the different results obtained in the batches, the feasibility of the methodology was tested using a commercial flow reactor of 1.2 mL (Lonza FlowPlate Lab used on an MMRS flow reactor from Ehrfeld). If compared to microwave irradiation technology, the use of a microfluidic device can lead to an easier scale-up, either by scaling up, sizing up or numbering up [69]. For this study, the system was designed to enable the continuous supply of a mixture containing either a PPI or a PAMAM precursor and carboxylic acid (PhA or FA) in methanol. The use of a back-pressure regulator (BPR) enabled us to work above the boiling point of the solvent, thereby accelerating the overall process. It is noteworthy that caffeic acid was not tested, as the different salts obtained tended to precipitate, leading to clogging of the system. The set-up of the system used for these syntheses is presented in Figure 3 below.

The high heat transfer provided by the microfluidic device, as well as this high temperature (100−120 °C) to perform the reaction within 5 min, as observed when microwave irradiation was used. This microwave/flow similarity is common and has been theorized to be the *microwave-to-flow paradigm* [70]. The different products were isolated as described in Section 3.1; in each case, yellowish solids were obtained, their purity being assessed by ^1^H NMR. The different yields, as well as the experimental conditions, are summarized in Table 4 below. Each test was duplicated.

### 2.3. Determination of the Antioxidant Properties

The antioxidant capacity of the ionic dendrimers was evaluated by measuring their ability to inhibit a DPPH radical. Several spectroscopic methods are described in the literature to measure the inhibition of the DPPH radical by an antioxidant [71], while UV-visible spectroscopy was used in another work [72]. The trapping of DPPH radicals by ionic dendrimers led to a color change from purple to yellow, relative to an H• transfer. This color change implies a decrease in the absorbance (A) of the DPPH radical at 515 nm. The results are expressed as a percentage of inhibition (I %), calculated according to the following formula:$$I\;\%=\frac{A\left(\mathrm{DPPH}\right)-A\left(\mathrm{DPPH}+\mathrm{antioxidant}\right)}{A\left(\mathrm{DPPH}\right)}\times 100$$
where A(DPPH) is the absorbance of DPPH and A(DPPH + antioxidant) is the absorbance of the mixture of DPPH and substrate at 515 nm.

The evaluation of the antioxidant power was determined for 5 families of dendrimers, namely, PPIs-FA, PPIs-CA, PAMAMs-FA, PAMAMs-CA and PAMAMs-PhA and caffeic and ferulic acids, with ascorbic acid (vitamin C) used as the positive control [73].

The effect of dendrimer concentration is shown in Figure 1 and Figure 2. From these curves, the IC_50_ of each dendrimer, i.e., the concentration of an antioxidant to inhibit 50 percent of free radicals at a given concentration, was graphically determined (Table 4). The IC_50_ curves in mg/L are given in the Appendix A.

These results show that antioxidant activity depends on the generation of the dendrimer (Table 4, entries 3–5, 6–8, 10–12 and 13–15). Indeed, for higher generations, more phenolic carboxylates are present, leading to higher antioxidant molar activities (Figure 1).

Dendrimers derived from PAMAMs exhibit higher antioxidant activity, compared to PPI-based ones for the same generation. Once again, this observation can be ascribed to a larger number of phenolic carboxylates for PAMAM, compared to PPI (a ratio of 2) for a given generation. The level of antioxidant activity also depends on the nature of the antioxidant and is mainly linked to the number of hydroxyl functions present in the structures. Indeed, the best antioxidant activities are obtained for the caffeic acid derivatives, PPI-3-CA and PAMAM3-CA, where the IC_50_ values are 3.5 ± 1.1 µmol/L and 1.0 ± 0.4 µmol/L (Figure 2), respectively, while the antioxidant activities of the ferulic acid derivatives are lower (Table 4, entries 3–8). Furthermore, it is important to mention that caffeic acid not only possesses better antioxidant properties but is also more stable than vitamin C [74,75].

As the dendrimers contain several antioxidant units, a ratio between ionic dendrimers and the corresponding acid was calculated (Table 4). Thus, we observed that ionic PPIs or PAMAM ferulates are more strongly antioxidant than pure ferulic acid, while in the case of caffeic acid, the latter remains the most antioxidant. This could be explained by a close interaction between OH groups of CA from the different arms of the dendrimer and/or OH and NH from the dendrimer itself; the solubility of the complex dendrimer/CA could also explain the difference in antioxidant activity.

In conclusion, dendrimers show good or even very good antioxidant activities and have the characteristic of being able to encapsulate various compounds of interest, particularly in the field of cosmetics.

For all ferulic and caffeic dendrimers, absorbance measurements at 515 nm were recorded every 90 s for 45 min and revealed no significant variation in absorbance as a function of time after 5 min of reaction with DPPH, meaning that their antioxidant activities appear quickly and are stable. On the other hand, a significant variation in the absorbance is observed for PAMAM-3-PhA with the passage of time (Figure 3). Indeed, after 5 min, the percentage of inhibition of 300 µM DPPH by PAMAM-3-PhA at 5 mM concentration is 45%, while it increases to more than 90% for t = 18 min (Figure 3). For this reaction time, the IC_50_ of PAMAM-3-PhA is 480 µmol/L. The antioxidant power of the latter is, thus, less important and takes more time to set up, compared to the other families of dendrimers derived from ferulic and caffeic acids.

### 2.4. Determination of Cytotoxic Properties

Based on the results of the toxicity tests performed on GD-PPI-4 and GD-PAMAM-3 dendrimers on human MRC5 fibroblasts [5,76], the cytotoxicity of five dendrimers, PPI-3-FA, PPI-3-PhA, PAMAM-3-FA, PAMAM-3-PhA, and PAMAM-3-CA was tested on human dermal fibroblasts using two methods, the WST1 assay and cell staining with crystal violet, as described in Section 3.1. Dermal fibroblasts were incubated for 48 h in the presence of increasing concentrations of each dendrimer, while the number of living cells was established by a WST-1 reduction test. Cell toxicity with all dendrimers was observed as soon as a concentration of 100 µg/mL was reached, except in the case of PPI-3-PhA, where toxicity was observed at 10 µg/mL (Figure 4a,b). At 100 µg/mL, 80%, 67%, 60%, 13% and 2% of living cells were detected in the presence of PAMAM-3-FA, PPI-3-FA, PAMAM-3-PhA, PAMAM-3-CA and PPI-3-PhA, respectively. Above this concentration, cell viability decreased with an increase in concentration to about 2% at 1000 µg/mL for all studied dendrimers. Dendrimers derived from ferulic acid remained less toxic than caffeic- and phloretic-derived ones, while the PPI-3-PhA dendrimer was the most toxic. Figure 5 shows the images of human dermal fibroblast cells in the presence of different dendrimers using an EVOS XL Core inverted microscope, whereby the cytotoxicity is well highlighted. Hence, at 100µg/mL, PAMAM-3-FA and PPI-3-FA could be used in cosmetics as encapsulating and antioxidant agents for active ingredients.

## 3. Conclusions

In the present work, 18 new ionic dendrimers have been easily prepared from PPI and PAMAM dendrimers and 3 biobased phenolic acids through a classical acid basic procedure. These syntheses were optimized under microwave activation and then transferred to a continuous flow microreactor, leading to high yields within short reaction times. The ionic dendrimers present very interesting antioxidant activities, even being superior to that of vitamin C, which was taken as the reference substance. These antioxidant activities depend on both the nature of the dendrimer and the phenolic acid used. The best antioxidant activities are observed with dendrimers PAMAMs-1-3-FA and PAMAMs-1-3-CA, both of these being stable compounds. Hence, at 100 µg/mL, PAMAM-3-FA and PPI-3-FA could be used in cosmetics as encapsulating and antioxidant agents for active ingredients. Studies of encapsulation of bioactive compounds are in progress and the synergy of the antioxidant power of the dendrimer and its encapsulation capacity are now under investigation.

### 3.1. Experimental

PPIs, PAMAMs, phloretic acid, ferulic acid, caffeic and solvent were purchased from traditional chemical suppliers and used without further purification. NMR spectra (d: doublet, t: triplet and qt: quintuplet) were recorded at 298 K at 500 MHz for ^1^H and 125 MHz for ^13^C on an Avance III Bruker spectrometer in CD_3_OD or (CD_3_)_2_SO. IR spectra (S: strong) were recorded on a Perkin Elmer Spectrum Two. Elemental analyses (C, H, N) were realized on a Flash EA-1112 Series. The microwave-assisted syntheses were performed on a CEM-focused microwave synthesis system in 20 mL vessels with septa at 90 Watts at 100 °C for 5 min. The absorbance measurements were realized on a FLUOstar Omega at 37 °C at 515 nm, taken every 90 s for 45 min.

#### 3.1.1. General Procedure for the Synthesis of PPI-PhA and PPI-FA Generations 1, 2 and 3

In a double-necked round-bottomed flask, phloretic acid (ferulic acid) (1.05 eq/NH_2_) and the dendrimer PPI-n (1 eq) were dissolved in methanol (20 mL). The reaction mixture was stirred at reflux (65 °C) for 5 h; then, methanol was evaporated under reduced pressure. The residue obtained thereby was then dissolved in water (20 mL) and extracted with ethyl acetate (20 mL). The aqueous phase was then washed twice with ethyl acetate (2 × 50 mL) to remove excess phloretic acid (ferulic acid) and then evaporated under reduced pressure. The yellow solid salts were obtained in quantitative yields of >97%.


Data for PPI-1-PhA:




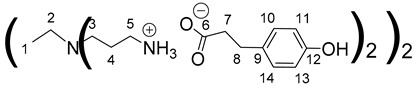



^1^H NMR (500 MHz, CD_3_OD), δ (ppm): 1.42–1.44 (4H, H1, m); 1.76–1.78 (8H, H4, m); 2.41–2.49 (12H, H7, H2, m); 2.54 (8H, H3, t, *J* = 6.98 Hz); 2.82 (8H, H8, t, *J* = 7.42 Hz); 2.91 (8H, H5, t, *J* = 7.24 Hz); 6.71 (8H, H11, H13, d, *J* = 8.61 Hz); 7.05 (8H, H10, H14, d, *J* = 8.61 Hz).

^13^C NMR (125 MHz, CD_3_OD), δ (ppm): 24.0 (C1); 24.2 (C4); 31.4 (C8); 38.2 (C5); 39.8 (C7); 50.9 (C3); 53.2 (C2); 114.9 (C11, C13); 128.9 (C10, C14); 132.9 (C9); 155.1 (C12); 180.4 (C6).

IR (cm^−1^): 1540 (νCO _carboxylate_, S), 1396 (νCO _carboxylate_, S), 1210–1235 (νC-O _carboxylate_, S).

Calculated elemental analysis of C_52_H_80_N_6_O_12_, 2H_2_O: C: 61.4%; H: 8.32%; N: 8.26%; O: 22.02%.

Experimental elemental analysis of C_52_H_80_N_6_O_12_, 2H_2_O: C: 61.79%; H: 8.56%; N: 8.58%; O: 21.06%.


Data for PPI-2-PhA:




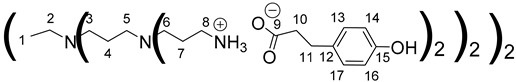



^1^H NMR (500 MHz, CD_3_OD), δ (ppm): 1.49–1.53 (4H, H1, m); 1.60–1.80 (24H, H4, H7, m); 2.39–2.69 (52H, H10, H2, H5, H3, H6, m); 2.80–2.83 (16H, H11, t, *J* = 7.42 Hz); 2,88 (16H, H8, t, *J* = 7.85 Hz); 6.70 (16H, H14, H16, d, *J* = 8.61 Hz); 7,05 (16H, H13, H17, d, *J* = 8.61 Hz).

^13^C NMR (125 MHz, CD_3_OD), δ (ppm): 22.41 (C4); 23.3 (C1); 24.6 (C7); 31.5 (C11); 38.1 (C8); 39.8 (C10); 50.7 (C6); 51.2 (C3); 51.3 (C5); 53.2 (C2); 114.8 (C14, C16); 128.9 (C13, C17); 132.9 (C12); 155.2 (C15); 180.5 (C9).

IR (cm^−1^): 1540 (νCO _carboxylate,_ S), 1396 (νCO _carboxylate_, S), 1210–1235 (νC-O _carboxylate_, S).

Calculated elemental analysis of C_112_H_176_N_14_O_24_, 5H_2_O: C: 61.35%; H: 8.55%; N: 8.94%; O: 21.16%.

Experimental elemental analysis of C_112_H_176_N_14_O_24_, 5H_2_O: C: 61.42%; H: 8.61%; N: 8.69%; O: 21.27%.


Data for PPI-3-PhA:




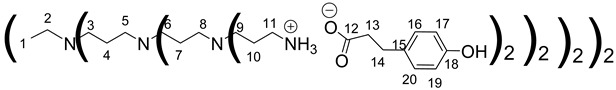



^1^H NMR (500 MHz, CD_3_OD), δ (ppm): 1.43–1;47 (4H, H1, m); 1.58–1.80 (56H, H4, H7, H10, m); 2.30–2.68 (116H, H13, H2, H5, H8, H3, H6, H9, m); 2.76–2.97 (64H, H14, H11, m); 6.72 (32H, H17, H19, d, *J* = 8.5 Hz); 7,05 (32H, H16, H20, d, *J* = 8.5 Hz).

^13^C NMR (125 MHz, CD_3_OD), δ (ppm): 22.4 (C4); 23.26 (C1); 24.6–24.6 (C7, C10); 31.5 (C14); 38.1 (C11); 39.8 (C13); 50.7 (C6, C9); 51.2 (C3); 51,3 (C5, C8); 53.2 (C2); 114.8 (C17, C19); 128.9 (C16, C20); 132.9 (C15); 155.2 (C18); 180.5 (C12)

IR (cm^−1^): 1540 (νCO _carboxylate,_ S), 1396 (νCO _carboxylate_, S), 1210–1235 (νC-O _carboxylate_, S).

Calculated elemental analysis of C_232_H_368_N_30_O_48_, 9H_2_O: C: 61.82%; H: 8.63%; N: 9.32%; O: 20.23%.

Experimental elemental analysis of C_232_H_368_N_30_O_48_, 9H_2_O: C: 62.14%; H: 8.62%; N: 9.32%; O: 20.40%.


Data for PPI-1-FA:




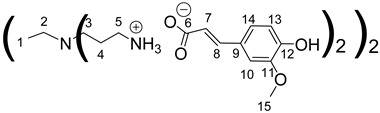



^1^H NMR (500 MHz, CD_3_OD), δ (ppm): 1.31–1.33 (4H, H1, m); 1.76–1.80 (8H, H4, m); 2.43–2.45 (4H, H2, m); 2.54 (8H, H3, t, *J* = 7.35 Hz); 2.98 (8H, H5, t, *J* = 7.35 Hz); 3.86 (12H, H15, s); 6.36 (4H, H7, d, *J* = 15.82 Hz); 6.82 (4H, H13, d, *J* = 8.11 Hz); 6.98 (4H, H14, dd, *J* = 1.90 Hz and *J* = 8.11 Hz); 7.09(4H, H10, d, *J* = 1.90 Hz); 7.35 (4H, H8, d, *J* = 15.82 Hz).

^13^C RMN (125 MHz, CD_3_OD), δ (ppm): 24.1 (C1); 24.8 (C4); 38.4 (C5); 51.0 (C3); 53.2 (C2); 55.0 (C15); 109.9 (C10); 115.0 (C13); 121.4 (C14); 121.8 (C7); 127.7 (C9); 140.3 (C8); 147.9 (C11); 147.9 (C12); 174.7 (C6).

IR (cm^−1^): 1634 (νCO _carboxylate,_ S), 1372 (νCO _carboxylate_, S), 1211 (νC-O _carboxylate_, S).

Calculated elemental analysis of C_56_H_84_N_6_O_16_, 2H_2_O: C: 59.56%; H: 7.50%; N: 7.40%; O: 25.5%.

Experimental elemental analysis of C_56_H_84_N_6_O_16_, 2H_2_O: C: 58.80%; H: 7.77%; N: 8.08%; O: 25.34%.


Data for PPI-2-FA:




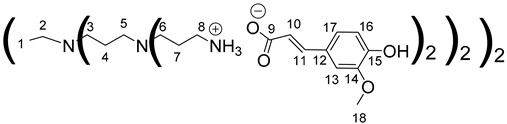



^1^H NMR (500 MHz, CD_3_OD), δ (ppm): 1.40–1.43 (4H, H1, m); 1.59–1.62 (8H, H4, m); 1.79 (16H, H7, m); 2.32–2.67 (36H, H2, H5, H3, H6, m); 2.94 (16H, H8, t, *J* = 7.08 Hz); 3.85 (24H, H18, s); 6.36 (8H, H10, d, *J* = 15.8 Hz); 6,81 (8H, H16, d, *J* = 8.13 Hz); 6,98 (8H, H17, dd, *J* = 8.13 Hz, *J* = 1.91 Hz); 7.09 (8H, H13, d, *J* = 1.91 Hz); 7.35 (8H, H11, d, *J* = 15.8 Hz).

^13^C NMR (125 MHz, CD_3_OD), δ (ppm): 22.4 (C4); 23.2 (C1); 24.6 (C7); 38.2 (C8); 50.8 (C6); 51.1 (C3); 51.2 (C5); 53.2 (C2); 55.1 (C18); 109.9 (C13); 115.3 (C16); 121.5 (C17); 121.9 (C10); 127.4 (C12); 140.3 (C11); 148.0 (C14); 148.3 (C15); 174.8 (C9).

IR (cm^−1^): 1635 (νCO _carboxylate,_ S), 1371 (νCO _carboxylate_, S), 1211 (νC-O _carboxylate_, S).

Calculated elemental analysis of C_120_H_176_N_14_O_32_, 5H_2_O: C: 59.64%; H: 7.76%; N: 8.11%; O: 24.49%.

Eexperimental elemental analysis of C_120_H_176_N_14_O_32_, 5H_2_O: C: 58.9%; H: 7.99%; N: 8.61%; O: 24.5%.


Data for PPI-3-FA:




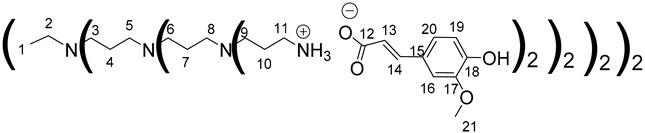



^1^H NMR (500 MHz, CD_3_OD), δ (ppm): 1.42–1.46 (4H, H1, m); 1.50–1.81 (56H, H4, H7, H10, m); 2.27–2.62 (84H, H2, H5, H8, H3, H6, H9, m); 2.92–2.96 (32H, H11, m); 3.87 (48H, H21, s); 6.35 (16H, H13, d, *J* = 15.83 Hz); 6.80 (16H, H19, d, *J* = 8.13 Hz); 6.98 (16H, H20, dd, *J* = 1.9 Hz and *J* = 8.13 Hz); 7.10 (16H, H16, d, *J* = 1.9 Hz); 7.34 (16H, H14, d, *J* = 15.83 Hz).

^13^C NMR (125 MHz, CD_3_OD), δ (ppm): 18.5 (C7); 22.9 (C4); 24,8 (C1, C10); 38.3 (C11); 50.8 (C6, C9, C3); 51.2 (C8, C5, C2); 55.1 (C21); 109.9 (C16); 115.3 (C19); 121.5 (C20); 121.9 (C13); 127.4 (C15); 140.2 (C14); 148.1 (C17); 148.1 (C18); 174.6 (C12).

IR (cm^−1^): 1635 (νCO _carboxylate,_ S), 1373 (νCO _carboxylate_, S), 1211 (νC-O _carboxylate_, S).

Calculated elemental analysis of C_248_H_368_N_30_O_64_, 13H_2_O: C: 59.24%; H: 7.9%; N: 8.36%; O: 24.5%.

Experimental elemental analysis of C_248_H_368_N_30_O_64_, 13H_2_O: C: 58.88%; H: 8.05%; N: 8.87%; O: 24.2%.

#### 3.1.2. General Procedure for the Microwave-Assisted Synthesis of PPI-PhA and PPI-FA Generations 1, 2 and 3

Phloretic acid (ferulic acid) (1.05 eq/NH_2_) and PPI-n dendrimer (1 eq), dissolved in methanol (7 mL), were introduced to a CEM microwave oven; the mixture was heated at 100 °C for 5 min at a maximal power of 90 W. The mixture was then transferred to a flask and methanol was evaporated under reduced pressure. The residue obtained thereby was then dissolved in water (20 mL) and extraction with ethyl acetate (20 mL) was performed. The aqueous phase was then washed twice with ethyl acetate (2 × 50 mL) to remove excess phloretic acid (ferulic acid) and evaporated under reduced pressure. The yellow solid salts were obtained with quantitative yields of >98%.

#### 3.1.3. General Procedure for the Flow Synthesis of Ionic Dendrimers

A mixture containing 2.0 g of PPI, the adequate number of carboxylic acid (1 eq/NH_2_) and methanol (33.3 g) was prepared in a flask and stirred at room temperature for about 1 min until a homogeneous liquid was obtained. The mixture was then pumped into the Ehrfeld MMRS system at 0.24 mL/min using a Knauer Azura P4.15 HPLC pump. The flow reactor is represented in Figure 3 and consists of a FlowPlate lab with a process plate LL, an insulation module and a backpressure regulator (Bronkhorst EL-Press). The temperature was settled to 150 °C using a Pilot One Ministat 230; a pressure of 3.5 bars was maintained via the BPR to prevent the evaporation of the medium. The residence time was confirmed by weighting the mixture obtained for a set amount of time (20 min). Each sample was then evaporated under vacuum at 45 °C, dissolved in methanol, washed three times with pentane/ethyl acetate (7/3) and dried under vacuum to obtain yellowish solids.

#### 3.1.4. General Procedure for the Synthesis of PPI-CA Generations 1, 2 and 3

Caffeic acid (1.1 eq/NH_2_), dissolved in the ethanol (15 mL), was added dropwise in a round-bottomed flask containing the PPI-n dendrimer (1 eq.) dissolved in ethanol (10 mL). The reaction mixture was stirred at room temperature for 2 h using orbital stirring. The mixture was transferred to a centrifuge tube; the precipitation was recovered and washed 3 times with ethanol (3 × 30 mL) to remove excess caffeic acid before being dried under reduced pressure. The yellow solid salts were obtained, with yields of 90%, 90% and 92%, respectively, for generations 1, 2 and 3.


Data for PPI-1-CA:




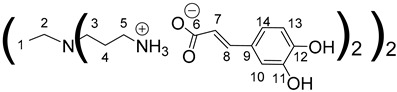



^1^H NMR (500 MHz, (CD_3_)_2_SO), δ (ppm): 1.30–1.34 (4H, H1, m); 1.60–1.64 (8H, H4, m); 2.27–2.30 (4H, H2, m); 2.37 (8H, H3, t, *J* = 6.87 Hz); 2.73 (8H, H5, t, *J* = 7.26 Hz); 6.14 (4H, H7, d, *J* = 15.76 Hz); 6.69 (4H, H13, d, *J* = 8.08 Hz); 6.80 (4H, H14, dd, *J* = 2.04 Hz and *J* = 8.08 Hz); 7,00 (4H, H10, d, *J* = 2.04 Hz); 7.13 (4H, H8, d, *J* = 15.76 Hz).

^13^C NMR (125 MHz, (CD_3_)_2_SO), δ (ppm): 24.9 (C1); 26.9 (C4); 38.6 (C5); 51.4 (C3); 53.6 (C2); 114.7 (C10); 116.3 (C13); 120.3 (C14); 122.4 (C7); 127.3 (C9); 140.1 (C8); 146.4 (C11); 147.9 (C12); 171.6 (C6).

IR (cm^−1^): 1504–1600 (νCO _carboxylate,_ S), 1375 (νCO _carboxylate_, S), 1204–1230 (νC-O _carboxylate_, S).

Calculated elemental analysis of C_52_H_72_N_6_O_16_, 2H_2_O: C: 58.2%; H: 7.14%; N: 7.83%; O: 26.83%.

Experimental elemental analysis of C_52_H_72_N_6_O_16_, 2H_2_O: C: 57.4%; H: 7.46%; N: 8.55%; O: 26.59%.


Data for PPI-2-CA:




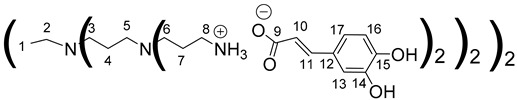



^1^H NMR (500 MHz, (CD_3_)_2_SO), δ (ppm): 1.27–1; 32 (4H, H1, m); 1.39–1.44 (8H, H4, m); 1.59–1.63 (16H, H7, m); 2.2–2.41 (36H, H2, H5, H3, H6, m); 2.74 (16H, H8, t, *J* = 6.95 Hz); 6.15 (8H, H10, d, *J* = 15.76 Hz); 6,71 (8H, H16, d, *J* = 8.09 Hz); 6.85 (8H, H17, dd, *J* = 8.09 Hz and *J* = 1.95 Hz); 7,00 (8H, H13, d, *J* = 1.95 Hz); 7.23 (8H, H11, d, *J* = 15.76 Hz).

^13^C NMR (125 MHz, (CD_3_)_2_SO), δ (ppm): 24.8 (C1); 26,8 (C4); 27.1 (C7); 38.6 (C8); 49.9 (C6); 50.1 (C3); 51.3 (C5); 53.1 (C2); 109.1 (C13); 114.8 (C16); 116.3 (C17); 120.7 (C10); 126.9 (C12); 141.9 (C11); 146.3 (C14); 148.2 (C15); 171.6 (C9).

IR (cm^−1^): 1504–1600 (νCO _carboxylate,_ S), 1375 (νCO _carboxylate_, S), 1204–1230 (νC-O _carboxylate_, S).

Calculated elemental analysis of C_112_H_160_N_14_O_32_, 5H_2_O: C: 58.37%; H: 7.72%; N: 9.15%; O: 25.39%.

Experimental elemental analysis of C_112_H_160_N_14_O_32_, 5H_2_O: C: 57.21%; H: 7.9%; N: 9.72%; O: 25.16%.


Data for PPI-3-CA:




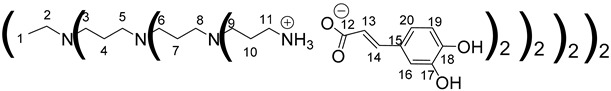



^1^H NMR (500 MHz, (CD_3_)_2_SO), δ (ppm): 1.25–1.73 (60H, H1, H4, H7, H10, m); 2.16–2.43 (84H, H2, H5, H8, H3, H6, H9, m); 2.74 (32H, H11, m); 6.15 (16H, H13, d, *J* = 15.75 Hz); 6.73 (16H, H19, d, *J* = 8.14 Hz); 6.87 (16H, H20, d, *J* = 8.14 Hz); 7.01 (16H, H16, s); 7.27 (16H, H14, d, *J* = 15.75 Hz).

^13^C NMR (125 MHz, (CD_3_)_2_SO), δ (ppm): 24.8 (C1); 26.1 (C4); 27.0 (C7, C10); 38.6 (C11); 49.8 (C6, C9, C3); 51.3 (C8, C5, C2); 111.7 (C16); 114.9 (C19); 116.2 (C20); 120.9 (C13); 126.7 (C15); 140.2 (C14); 146.4 (C17); 148.0 (C18); 170.7 (C12).

IR (cm^−1^): 1504–1600 (νCO _carboxylate,_ S), 1375 (νCO _carboxylate_, S), 1204–1230 (νC-O _carboxylate_, S).

Calculated elemental analysis of C_196_H_316_N_30_O_54_, 7H_2_O: C: 57.66%; H: 8.15%; N: 10.29%; O: 23.9%.

Experimental elemental analysis of C_196_H_316_N_30_O_54_.7H_2_O: C: 57.05%; H: 8.26%; N: 10.75%; O: 23.94%.

#### 3.1.5. General Procedure for the Synthesis of PAMAM-PhA and PAMAM-FA Generations 1, 2 and 3

Phloretic acid (ferulic acid) (1.5 equiv. per NH_2_) and PAMAM-n dendrimer (1 equiv.), dissolved in methanol (20 mL), were introduced to a double-necked round-bottomed flask. The mixture was stirred at reflux (65 °C) for 16 h and the methanol was evaporated under reduced pressure. The residue obtained thereby was then dissolved in water (20 mL) and extracted with ethyl acetate (20 mL). The aqueous phase was then washed twice with ethyl acetate (2 × 50 mL) to remove the excess phloretic acid (ferulic acid). After evaporation, the yellow solid salts were obtained, with quantitative yields of >98%.


Data for PAMAM-1-PhA:




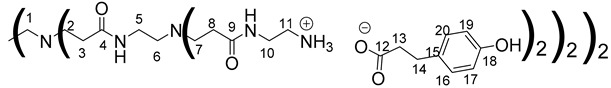



^1^H NMR (500 MHz, CD_3_OD), δ (ppm): 2.29–2.49 (40H, H3, H8, H13, m); 2.5–2.61 (12H, H1, H6, m); 2.64–2.95 (40H, H2, H7, H14, m); 2.97–3.1 (16H, H11, m); 3.26 (8H, H5, m); 3.47–3.49 (16H, H10, m); 6.72 (16H, H16, H20, d, *J* = 8.46 Hz) 7.05 (16H, H17, H19, d, *J* = 8.46 Hz).

^13^C NMR (125 MHz, CD_3_OD), δ (ppm): 31.5 (C14); 33.1 (C3, C8); 37.3 (C5, C10); 39.3 (C11); 39.8 (C13); 49.5 (C2, C7); 52.0 (C1, C6); 114.9 (C16, C20); 128.9 (C17, C19); 132.8 (C15); 155.1 (C18); 173.3 (C4); 174.4 (C9); 180.6 (C12).

IR (cm^−1^): 1540 (νCO _carboxylate,_ S), 1394 (νCO _carboxylate_, S), 1211–1236 (νC-O _carboxylate_, S).

Calculated elemental analysis of C_134_H_208_N_26_O_36._13H_2_O: C: 53.77%; H: 7.88%; N: 12.17%; O: 26.19%.

Experimental elemental analysis of C_134_H_208_N_26_O_36._13H_2_O: C: 54.1%; H: 8.32%; N: 11.34%; O: 26.23%.


Data for PAMAM-2-PhA:




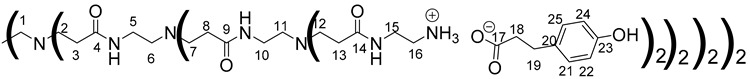



^1^H NMR (500 MHz, CD_3_OD), δ(ppm): 2.29–2.50 (88H, H 3, H8, H13, H18, m); 2.51–2.63 (28H, H1, H6, H11, m); 2.67–2.92 (88H, H2, H7, H12, H19, m); 2.94–3.1 (32H, H16, m); 3.26 (24H, H5, H10, m); 3.38–3.48 (32H, H15, m); 6.72 (32H, H16, H20, d, *J* = 8.27 Hz) 7.05 (32H, H17, H19, d, *J* = 8.27 Hz);

^13^C NMR (125 MHz, CD_3_OD), δ (ppm): 31.52 (C19); 33.1 (C3, C8, C13); 37.3 (C5, C10, C15); 39.3 (C16); 39.8 (C18); 49.5 (C2, C7, C12); 52.0 (C1, C6, C11); 114.9 (C21, C25); 128.9 (C22, C24); 132.8 (C20); 155.1 (C23); 173.3 (C4, C9); 174.4 (C14); 180.6 (C17).

IR (cm^−1^): 1540 (νCO _carboxylate,_ S), 1394 (νCO _carboxylate_, S), 1211–1236 (νC-O _carboxylate_, S).

Calculated elemental analysis of C_286_H_448_N_58_O_76._32H_2_O: C: 52.92%; H: 7.95%; N: 12.51%; O: 26.62%.

Experimental elemental analysis of C_286_H_448_N_58_O_76._32H_2_O: C: 53.24%; H: 7.99%; N: 11.75%; O: 27.01%.


Data for PAMAM-3-PhA:




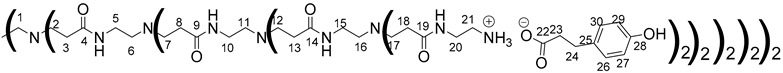



^1^H NMR (500 MHz, CD_3_OD), δ (ppm): 2.25–2.49 (184H, H3, H8, H13, H18, H23, m); 2.50–2.62 (60H, H1, H6, H11, H16, m); 2.7–2.89 (184H, H2, H7, H12, H17, H24, m); 2.98–3.09 (64H, H21, m); 3.19–3.30 (56H, H5, H10, H15, m); 3.44 (64H, H20, m); 6.71 (64H, H26, H30, d, *J* = 8.3 Hz); 7,05 (64H, H27, H29, d, *J* = 8.3 Hz).

^13^C NMR (125 MHz, CD_3_OD), δ (ppm): 31.5 (C24); 33.1 (C3, C8, C13); 33.3 (C18); 37.1–37.4 (C5, C10, C15, C20); 39.3 (C21); 39.8 (C23); 49.5 (C2, C7, C12, C17); 52.0 (C1, C6, C11, C16); 114.9 (C26, C30); 128.9 (C27, C29); 132.8 (C25); 155.1 (C28); 173.3 (C4, C9, C14); 174.4 (C19); 180.6 (C22).

IR (cm^−1^): 1540 (νCO _carboxylate,_ S), 1394 (νCO _carboxylate_, S), 1211–1236 (νC-O _carboxylate_, S).

Calculated elemental analysis of C_590_H_926_N_122_O_156._98H_2_O: C: 50.65%; H: 8.08%; N: 12.21%; O: 29.05%.

Experimental elemental analysis of C_590_H_926_N_122_O_156._98H_2_O: C: 51.04%; H: 8.00%; N: 11.40%; O: 29.04%.


Data for PAMAM-1-FA:




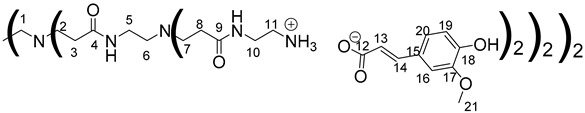



^1^H NMR (500 MHz, CD_3_OD), δ (ppm): 2.29–2.43 (24H, H3, H8, m); 2.48–2.61 (12H, H1, H6, m); 2.64–2.87 (24H, H2, H7, m); 3–3.13 (16H, H11, m); 3.23 (8H, H5, m); 3.47 (16H, H10, m); 3.86 (24H, H21, s) 6,38 (8H, H13, d, *J* = 15.85 Hz); 6.82 (8H, H19, d, *J* = 8.26 Hz); 6,99(8H, H20, dd, *J* = 1.95 Hz, *J* = 8.26 Hz); 7.11 (8H, H16, d, *J* = 1.95 Hz); 7.35 (8H, H14, d, *J* = 15.85 Hz).

^13^C NMR (125 MHz, CD_3_OD), δ (ppm): 33.2 (C3, C8); 37.2 (C5); 38.3 (C10); 39.7 (C11); 49.6 (C2, C7); 50.8 (C1); 52.0 (C6); 55.0 (C21); 109.9 (C16); 115.0 (C19); 121.5 (C13); 121.7 (C20); 127.7 (C15); 140.4 (C14); 147.9 (C17, C18); 173.2 (C4); 174.3 (C9); 174.6 (C12).

IR (cm^−1^): 1511–1590 (νCO _carboxylate,_ S), 1373 (νCO _carboxylate_, S), 1220–1264 (νC-O _carboxylate_, S).

Calculated elemental analysis of C_142_H_208_N_26_O_44._16H_2_O: C: 52.13%; H: 7.39%; N: 11.13%; O: 29.34%.

Experimental elemental analysis of C_142_H_208_N_26_O_44._16H_2_O: C: 52.21%; H: 7.79%; N: 11.13%; O: 28.87%.


Data for PAMAM-2-FA:




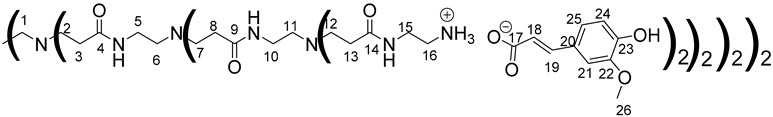



^1^H NMR (500 MHz, CD_3_OD), δ (ppm): 2.20–2.46 (56H, H3, H8, H13, m); 2.48–2.60 (28H, H1, H6, H11, m); 2.66–2.92 (56H, H2, H7, H12, m); 3.00–3.16 (32H, H16, m); 3.23 (24H, H5, H10, m); 3.50 (32H, H15, m); 3.86 (48H, H26, s) 6.37 (16H, H18, d, *J* = 15.8 Hz); 6.81 (16H, H24, d, *J* = 8.2 Hz); 6.99 (16H, H25, dd, *J* = 1.95 Hz, *J* = 8.2 Hz); 7.11 (16H, H21, d, *J* = 1.95 Hz); 7.35 (16H, H19, d, *J* = 15.8 Hz).

^13^C NMR (125 MHz, CD_3_OD), δ (ppm): 33.2 (C3, C8, C13); 37.5 (C5, C10); 38.3 (C15); 39.7 (C16); 49.6 (C2, C7, C12); 51.0 (C1, C6, C11); 55.0 (C26); 109.9 (C21); 115.0 (C24); 121.5 (C18, C25); 127.7 (C20); 140.4 (C19); 147.9 (C22, C23); 174.2 (C4, C9, C14); 174.6 (C17).

IR (cm^−1^): 1511–1590 (νCO _carboxylate,_ S), 1373 (νCO _carboxylate_, S), 1220–1264 (νC-O _carboxylate_, S).

Calculated elemental analysis of C_302_H_448_N_58_O_92._40H_2_O: C: 51.21%; H: 7.51%; N: 11.47%; O: 29.81%.

Experimental elemental analysis of C_302_H_448_N_58_O_92._40H_2_O: C: 51.53%; H: 7.68%; N: 10.93%; O: 29.84%.


Data for PAMAM-3-FA:




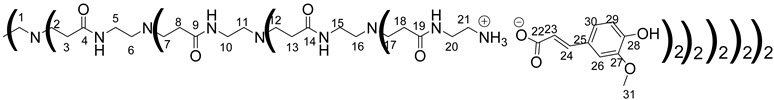



^1^H NMR (500 MHz, CD_3_OD), δ (ppm): 2.23–2.48 (120H, H3, H8, H13, H18, m); 2.47–2.61 (60H, H1, H6, H11, H16, m); 2.63–2.95 (120H, H2, H7, H12, H17, m); 3.02–3.17 (64H, H21, m); 3.17–3.28 (56H, H5, H10, H15, m); 3.40–3.56 (64H, H20, m); 3.86 (96H, H31, s); 6.37 (32H, H23, d, *J* = 15.8 Hz); 6.82 (32H, H29, d, *J* = 8.27 Hz); 6.99 (32H, H30, dd, *J* = 1.95 Hz, *J* = 8.27 Hz); 7.11 (32H, H26, d, *J* = 1.95 Hz); 7.36 (32H, H24, d, *J* = 15.8 Hz).

^13^C NMR (125 MHz, CD_3_OD), δ (ppm): 33.2 (C3, C8, C13, C18); 37.17 (C5, C10, C15); 38.0 (C20); 39.6 (C21); 49.5 (C2, C7, C12, C17); 51.9 (C1, C6, C11, C16); 55.16 (C31); 110.0 (C26); 115.3 (C29); 121.5 (C30); 122.0 (C23); 127.6 (C25); 140.4 (C24); 148.0 (C27, C28); 174.4 (C4, C9, C14, C19); 174.9 (C22).

IR (cm^−1^): 1511–1590 (νCO _carboxylate,_ S), 1373 (νCO _carboxylate_, S), 1220–1264 (νC-O _carboxylate_, S).

Calculated elemental analysis of C_622_H_928_N_122_O_188._95H_2_O: C: 50.36%; H: 7.60%; N: 11.52%; O: 30.52%.

Experimental elemental analysis of C_622_H_928_N_122_O_188._95H_2_O: C: 50.24%; H: 7.61%; N: 10.94%; O: 29.82%.

#### 3.1.6. General Procedure for the Synthesis of PAMAM-CA Generations 1, 2 and 3

Caffeic acid (1.5 equiv. per NH_2_), dissolved in ethanol (15 mL), was introduced dropwise to a round-bottomed flask containing the PAMAM-n dendrimer (1 eq.), which had been previously dissolved in ethanol (10 mL). The reaction mixture was stirred at room temperature for 20 h using orbital stirring. The mixture was then transferred to a centrifuge. The residue recovered by filtration was then washed 3 times with ethanol (3 × 30 mL) to remove the excess caffeic acid. The product obtained thereby after evaporation under reduced pressure consisted of a white solid salt, with quantitative yields of >99% for generations 1 and 3 and 90% for generation 2.


Data for PAMAM-1-CA:




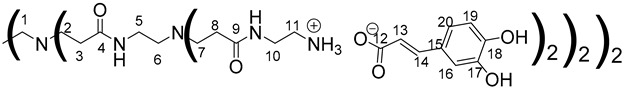



^1^H NMR (500 MHz, (CD_3_)_2_SO), δ (ppm): 2.14–2.99 (24H, H3, H8, m); 2.36–2.46 (12H, H1, H6, m); 2.56–2.7 (24H, H2, H7, m); 2.72–2.81 (16H, H11, m); 3.06–3.15 (8H, H5, m); 3.23 (16H, H10, m); 6.15 (8H, H13, d, *J* = 15.79 Hz); 6.71 (8H, H19, d, *J* = 8.22 Hz); 6.83 (8H, H20, dd, *J* = 2.08 Hz, *J* = 8.22 Hz); 6.98 (8H, H16, d, *J* = 2.08 Hz); 7.20 (8H, H14, d, *J* = 15.79 Hz).

^13^C NMR (125 MHz, (CD_3_)_2_SO), δ (ppm): 33.8 (C3, C8); 37.4 (C5, C10); 39.0 (C11); 50.0 (C2, C7); 52.6 (C1, C6); 114.8 (C16); 116.3 (C19); 120.6 (C13); 121.0 (C20); 127.1 (C15); 141.1 (C14); 146.3 (C17); 148.0 (C18); 171.2 (C4, C9); 172.5 (C12).

IR (cm^−1^): 1522–1580 (νCO _carboxylate,_ S), 1376 (νCO _carboxylate_, S), 1200–1220 (νC-O _carboxylate_, S).

Calculated elemental analysis of C_134_H_192_N_26_O_44_.11H_2_O: C: 52.44%; H: 7.03%; N: 11.87%; O: 28.67%.

Experimental elemental analysis of C_134_H_192_N_26_O_44._11H_2_O: C: 52.64%; H: 7.41%; N: 11.63%; O: 28.31%.


Data for PAMAM-2-CA:




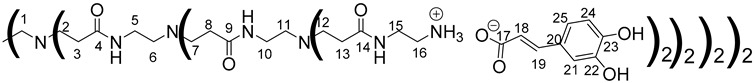



^1^H NMR (500 MHz, (CD_3_)_2_SO)), δ (ppm): 2.02–2.29 (56H, H3, H8, H13, m); 2.34–2.47 (28H, H1, H6, H11, m); 2.56–2.79 (88H, H2, H7, H12, H15,m); 2.99- 3.24 (56H, H5, H10, H15, m); 6.15 (13H, H18, d, *J* = 15.74 Hz); 6.73 (13H, H24, d, *J* = 8.17 Hz); 6.86 (13H, H25, dd, *J* = 2.14 Hz, *J* = 8.17 Hz); 6.98 (13H, H21, d, *J* = 2.14 Hz); 7.26 (13H, H19, d, *J* = 15.74 Hz).

^13^C NMR (125 MHz, CD_3_OD), δ (ppm): 33.7 (C3, C8, C13); 37.4 (C5, C10, C15); 39.0 (C16); 50.0 (C2, C7, C12); 52.5 (C1, C6, C11); 114.7 (C21); 116.3 (C24); 120.6 (C23); 121.0 (C25); 127.0 (C18); 141.1 (C19); 146.2 (C22, C23); 171.2 (C4, C9, C14); 172.7 (C17).

IR (cm ^−1^): 1522–1580 (νCO _carboxylate,_ S), 1376 (νCO _carboxylate_, S), 1200–1220 (νC-O _carboxylate_, S).

Calculated elemental analysis of C_286_H_416_N_58_O_92._20H_2_O: C: 52.86%; H: 7.07%; N: 12.50%; O: 27.57%.

Experimental elemental analysis of C_286_H_416_N_58_O_92._20H_2_O: C: 52.21%; H: 7.09%; N: 12.87%; O: 27.83%.


Data for PAMAM-3-CA:




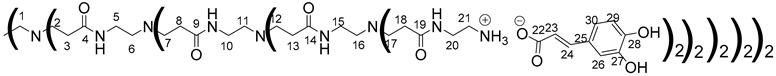



^1^H NMR (500 MHz, (CD_3_)_2_SO), δ (ppm): 2.10–2.34 (120H, H3, H8, H13, H18, m); 2.35–2.48 (60H, H1, H6, H11, H16, m); 2.55–2.87 (184H, H2, H7, H12, H17, H21, m); 2.95–3.33 (120H, H5, H10, H15, H20, m); 6,16 (32H, H23, d, *J* = 15.79 Hz); 6.72 (32H, H29, d, *J* = 8.24 Hz); 6.85 (32H, H30, dd, *J* = 2.03 Hz, and *J* = 8.24 Hz); 6.98 (32H, H26, d, *J* = 2.03 Hz); 7.24 (32H, H24, d, *J* = 15.79 Hz).

^13^C NMR (125 MHz, (CD_3_)_2_SO), δ (ppm): 33.8 (C3, C8, C13, C18); 37.4 (C5, C10, C15, C20); 39.1 (C21); 50.0 (C2, C7, C12, C17); 52.6 (C1, C6, C11, C16); 114.8 (C26); 116.4 (C29); 120.6 (C23); 121.1 (C30); 127.1 (C25); 141.1 (C24); 146.3 (C27, C28); 171.2 (C4, C9, C14, C19); 172.6 (C22).

IR (cm^−1^): 1522–1580 (νCO _carboxylate,_ S), 1376 (νCO _carboxylate_, S), 1200–1220 (νC-O _carboxylate_, S).

Calculated elemental analysis of C_590_H_863_N_122_O_188._26H_2_O: C: 53.92%; H: 7.02%; N: 13.00%; O: 26.05%.

Experimental elemental analysis of C_590_H_863_N_122_O_188._26H_2_O: C: 53.16%; H: 7.50%; N: 13.75%; O: 25.58%.

#### 3.1.7. Antioxidant Activity Evaluation of Synthesized Dendrimers

In a volumetric flask, a solution of 2,2-diphenyl 1-picrylhydrazyl (DPPH•) at 6 × 10^−4^ mol/L was prepared (2.37 mg in 25 mL of methanol). Then, a solution of the desired antioxidant ionic dendrimers was prepared, either in methanol for the phloretic and ferulic acid-based dendrimers or in a methanol/water/DMSO ((3:1:1) *v*/*v*/*v*) mixture for the caffeic acid-based dendrimers. The solution was diluted to different concentrations in microplate wells. An equivalent volume of DPPH• solution (100 µL/100 µL) was added to each well.

The absorbance measurements were realized using a FLUOstar Omega device at 515 nm every 90 s for 45 min to control the kinetics of the reaction. The curve of the percentage inhibition as a function of the concentrations (I (%) = f ([C])) gave the IC_50_ values (the concentration of the antioxidant necessary to inhibit 50% of DPPH) for each compound.

### 3.2. Cytotoxicity Experiment

#### 3.2.1. Cell Culture

Normal human dermal fibroblasts were purchased from Promocell (Heidelberg, Germany). They were grown in DMEM, supplemented with 10% fetal bovine serum (FBS) according to the manufacturer’s specifications, in Nunclon^®^ 75 cm^2^ flasks (Dutscher Brumath, France) at 37 °C in a humid atmosphere containing 5% CO_2_.

#### 3.2.2. Cell Viability Assay

Cells were seeded in sterile 96-well microtiter plates (1 × 10,000 cells/well) and were allowed to settle for 24 h. Dendrimers were added to the cells at final concentrations of 0, 1, 10, 200, 500 and 1000 µg/mL in DMEM supplemented with 1% FBS. The cells were incubated for 48 h and imaged using an EVOS XL Core (Invitrogen) inverted microscope (×10 magnification) (Figure 5). Then, the medium was replaced by fresh medium containing 10% Wst-1 reagent. Absorbance was measured at 450 nm using a microplate spectrophotometer (SPECTROstar^®^ Nano, BMG Labtech, Champigny-sur-Marne, France). Statistical analysis was performed using Student’s *t*-test. The results are expressed as the mean ± standard deviation.

## Data Availability

Not applicable.

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
