# Peer review of "Synthesis and Activity of Ionic Antioxidant-Functionalized PAMAMs and PPIs Dendrimers"

_polymers, 2022, doi:10.3390/polym14173513_

Round 1

Reviewer 1 Report

This paper prepared dendrimers using acid-base reaction with 3 different kinds of phenolic acids/ The synthesis with microwave and micro fluidic reactors was discussed. The applications for antioxidant, cytotoxic were characterized and discussed. However, there are some problems within this draft:

  1. The introduction should be rewritten. Currently it lacks novelty and impact.
  2. Table 1 has so many structures and it seems redundant. I would recommend the authors either move it to supporting information or redesign the table.
  3. A key problem within this draft is that the authors claimed the NMR and IR supported the argument of dendrimer synthesis. However, no spectrum was provided for either NMR or IR. Please add the spectrum to the supporting information. 
  4. Following on last point, no detailed discussions or information was found regarding the completion of base-acid reaction. I would assume it would be very difficult to fully react all the functional groups, particularly given only slightly excess (1.05 : 1) and higher generations. In other words, the real structures of generation 2 and 3 might be very different from the designed structures. Please comment on that and provide supporting information of the reaction completion.
  5. Following on point 4, the authors should try to use zeta potential to analyze the synthesized material properties. 
  6. Unless the authors provide enough supporting information about the purity of target dendrimers, the yield reported did not really provide any useful information. 
  7. Between line 196-197, the authors claimed the reaction temperature varied from 100C to 120C. However, in table 2, only data for 100C was provided. Please comment on that.
  8. Have the authors consider the following comparison experiment for cytotoxic determination: “use PPI/PAMAM + Acid to see their cytotoxic” to see the difference with the corresponding dendrimers? Please comment on that.

In addition, here are some other minor issues with this draft:

  1. For Scheme 1, (b) MW, 100C, 5min, MeOH, does “MW” microwave? Please make sure all the abbreviations are clearly defined before being used
  2. The qualities for Figure 1-2 are very low. Please improve that
  3. On line 266, it should be “On Figure 5 are” instead of “On figure 5”. Please make sure all the formats are correct within the draft
  4. Line 218, the authors said “The effect of dendrimer concentration is shown in Graphs 1 and 2.” It should be “Figure 1 and 2”.
  5. In Figure 3, it should be “6.5”, “9.5” and etc instead of “6,5”, “9,5” and etc.
  6. There are some other small issues within the draft. Please make sure all the formats, spellings and grammars are correct within the draft. 

Author Response

Dear Reviewer

thank you for your useful remarks and corrections.

You will find our answers in the attached document.

Sincerly

Reviewer 2 Report

The reviewer made comments in an attachment PDF.  Please find it.

Author Response

Dear reviewer

thanks you for your appropriate comments and questions. We have carefully revised the manuscript. You will find our answers in the attached document.

Kindly regards

Round 2

Reviewer 1 Report

Thank you for the revisions and responses.

The authors have updated the draft accordingly. However, here are still some minor changes need to be made.

  1. Still the introduction part. Thanks for the revision. However, current vision is still very hard for readers to follow. For example, from the end of the first paragraph to the beginning of second paragraph, the authors described “To our knowledge, ionic dendrimers or dendrimers presenting antioxidants properties have been described in the literature but no dendrimers presenting the two characteristics.   Indeed, the synthesis of ionic dendrimers has previously been described in the literature…” It lacks a clear flow/logic. Also, proper citations should be added for the revised paragraph. 
  2. In the supporting information, “MeOD” is not the right format. It should be CD3OD.
  3. The IR results qualities provided in the supporting information are very low. Currently they are just screenshots of the spectrums. The authors should try to improve their qualities by re-plot the data by themselves.

Author Response

The authors thank the reviewer for its work and replied to the remarks or questions.

Author Response

(The authors gave the same response as above.)
